# Effects of a Novel Mammalian-Derived Collagen Matrix on Human Articular Cartilage-Derived Chondrocytes from Osteoarthritis Patients

**DOI:** 10.3390/ijms26167826

**Published:** 2025-08-13

**Authors:** Mingyuan Wang, Toru Iwahashi, Taisuke Kasuya, Mai Konishi, Katsuyuki Konishi, Miki Kawanaka, Takashi Kanamoto, Hiroyuki Tanaka, Ken Nakata

**Affiliations:** 1Department of Sports Medical Science, Graduate School of Medicine, The University of Osaka, Osaka 565-0871, Japan; doctorwang9506@gmail.com (M.W.); miki.link0413@gmail.com (M.K.); 2Department of Medicine for Sports and Performing Arts, Graduate School of Medicine, The University of Osaka, Osaka 565-0871, Japan; takanamoto@hss.osaka-u.ac.jp (T.K.); ken.nakata@hss.osaka-u.ac.jp (K.N.); 3Department of Orthopaedic Surgery, Graduate School of Medicine, The University of Osaka, Osaka 565-0871, Japan; kurobuchi0918@ort.med.osaka-u.ac.jp (T.I.); kasuminn0623@gmail.com (T.K.); maik0112@aol.com (M.K.); kasyu524@gmail.com (K.K.)

**Keywords:** osteoarthritis, autologous chondrocyte implantation, chondrocytes, collagen, proteoglycans

## Abstract

Osteoarthritis (OA) is the most common joint disorder worldwide. Autologous chondrocyte implantation (ACI) is an established treatment for articular cartilage defects of the knee, but its effectiveness in OA is still under investigation. In this study, we investigated the effects of a newly developed mammalian-derived collagen matrix, NC-Col, on the proliferation, migration, adhesion, and gene expression of human articular cartilage-derived chondrocytes from OA patients in vitro, using proliferation assays, wound healing assays, adhesion assays, RT-qPCR, and RNA sequencing, respectively. In addition, the effects of NC-Col were compared with three different commercially available collagen matrices, and the underlying molecular mechanisms through which NC-Col influences these cellular behaviours were explored. Our results showed that NC-Col, used as a coating matrix, enhances cell proliferation, maintains the phenotype, and upregulates *Proteoglycan 4* (*PRG4*) in human articular cartilage-derived chondrocytes. Inhibition of the PI3K-Akt signalling pathway was found to be involved in some of these effects. In conclusion, our findings suggest that NC-Col collagen may offer new strategies for improving therapeutic outcomes in OA, particularly in the context of ACI.

## 1. Introduction

Osteoarthritis (OA), the most common degenerative joint disease, is characterised by cartilage degeneration and affects an estimated 250 million individuals worldwide [1]. Due to the combined impact of population ageing, the rising incidence of obesity, and an increasing number of joint injuries, the burden of OA continues to rise globally. It is reported that approximately one-third of individuals aged over 65 are affected by OA [2].

Knee osteoarthritis, the most common type of OA, is typically diagnosed around age 55, with treatment traditionally focusing on pain management until joint replacement becomes necessary after 10 to 15 years of disease progression [1,3]. Given that pain is the dominant symptom of OA, early-stage treatment commonly focuses on symptom relief. Education, lifestyle interventions, such as exercise, and weight management are also key non-surgical treatments [4]. Surgical interventions for knee osteoarthritis include knee joint distraction, knee osteotomy, arthroscopic knee surgery, and joint replacement for end-stage osteoarthritis. However, even arthroscopy, despite being widely performed, has been associated with an increased risk of earlier joint replacement, with its therapeutic effects often regarded as offering little more than a placebo effect [5].

OA is currently widely acknowledged as a whole-joint disorder involving structural changes across cartilage, bone, ligaments, synovium, capsule, and periarticular muscles. The development, homeostasis, and degeneration of cartilage are tightly associated with OA [6]. Articular cartilage, composed solely of chondrocytes embedded in a water-rich extracellular matrix rich in aggrecan and collagen II, has very limited capacity for self-repair [7]. Therefore, articular cartilage injury is a key risk factor for OA. To date, some researchers have investigated the interaction between collagen and chondrocytes in vitro and in vivo. According to previous studies, collagen matrices affect cell proliferation, adhesion, differentiation, gene expression, and metabolic activity in cartilage-derived chondrocytes [8,9,10]. In addition, collagen matrices were applied in autologous chondrocyte implantation (ACI) for the treatment of articular cartilage disorders [11].

ACI was first performed in Sweden in 1987 to treat a knee injury, with Brittberg and Peterson’s team later publishing the outcomes in 1994 [12]. It is one of the current methods for treating defects in the articular surface in the clinic. In this technique, chondrocytes are harvested from the patient, isolated, proliferated in culture, and then reimplanted autologously [8,13]. The procedure of ACI has evolved over time from original ACI to second-generation ACI and third-generation ACI [8]. In the first and second generations, cultured autologous chondrocytes were injected into chondral defects under a periosteal patch or a synthetic periosteal patch, respectively [8,13]. The third-generation ACI, known as matrix-induced autologous chondrocyte implantation (MACI), utilises a chondrocyte-seeded collagen membrane for repair [14]. Specifically, chondrocytes were seeded and cultured on the membrane in vitro for approximately four weeks prior to implantation, after which autologous chondrocytes were delivered to the defect site by the collagen membrane [15,16]. While continuously evolving techniques appear to improve the healing of cartilage defects, the problem of de-differentiation in chondrocytes in cell culture, which is characterised by a significantly reduced level of proteoglycan production and an increase in collagen I, still remains to be explored and solved [9,17]. Although ACI is not generally recommended as a standard treatment for OA, a meta-analysis published in 2023 reported that both ACI and MACI achieved long-term success and delayed arthroplasty in osteoarthritic knees with cartilage defects, however, the strength of this evidence is limited by small sample sizes and study heterogeneity [18]. In addition, a three-year follow-up study demonstrated that MACI significantly improved knee function and pain, and higher quality cartilage repair tissue was associated with reduced progression of patellofemoral osteoarthritis [19].

Recently, a newly developed mammalian-derived collagen matrix was developed by Saraya Co., Ltd. Our preliminary studies demonstrated that human articular cartilage-derived chondrocytes from OA patients cultured on the cell dish coated with this novel collagen matrix showed a significant upregulation of *Proteoglycan 4* (*PRG4*) expression. This finding suggests that the collagen holds potential as a novel material for OA treatment, particularly in the context of ACI. As such, in the present study, we aimed to evaluate the effects of a newly developed mammalian-derived collagen matrix, referred to as NC-Col, on cell proliferation, migration, adhesion, and gene expression of human articular cartilage-derived chondrocytes in vitro. Specifically, we investigated cell activities through proliferation assays, wound healing assays, and adhesion assays, and analysed gene expression using RT-qPCR and RNA sequencing. Furthermore, we also explored the underlying molecular mechanisms. The results showed that NC-Col could better enhance cell proliferation, maintain the phenotype, and upregulate *PRG4* expression in human articular cartilage-derived chondrocytes.

## 2. Results

### 2.1. NC-Col Promoted the Proliferation, Inhibited the Migration, and Enhanced the Adhesion Ability of Human Articular Cartilage-Derived Chondrocytes

To evaluate the effects of NC-Col on chondrocyte proliferation, we performed the CCK-8 assay at 24, 72, and 120 h. The results showed that NC-Col significantly promoted chondrocyte proliferation compared to the control and the porcine-derived collagen matrix (PDC) groups (*p* < 0.001) (Figure 1). Next, to assess its influence on cell migration, the wound healing assay was conducted at 12 and 24 h (Figure 2). The migration rate in the NC-Col group was markedly lower at two time points. At 12 h, the migration rate was 3.00% ± 0.46%, significantly lower than those of the control (13.73% ± 3.65%), fish-derived collagen matrix (FDC) (10.30% ± 0.96%), and PDC (13.87% ± 1.60%) groups. At 24 h, the NC-Col group maintained a reduced migration rate (5.97% ± 1.00%) compared to the control (30.60% ± 5.15%), FDC (20.27% ± 1.88%), and PDC (24.60% ± 1.83%) groups. Moreover, to investigate the effects of NC-Col on the adhesion ability of chondrocytes, we performed the adhesion assay. The cell adhesion assay revealed that NC-Col markedly enhanced the adhesion ability of chondrocytes. Notably, the adhesion-promoting effects of NC-Col, which were comparable to those observed with the PDC group, were significantly greater than those of the control and the bovine-derived collagen matrix (BDC) groups (*p* < 0.001) (Figure 3). Collectively, these results suggest that NC-Col effectively promotes the proliferation and adhesion of human articular cartilage-derived chondrocytes while significantly inhibiting their migration, compared to control, FDC, and PDC groups.

### 2.2. NC-Col Upregulated the Expression of ITGA2, ITGA10, and PRG4 in Chondrocytes

To evaluate the effects of the NC-Col on the gene expression of chondrocytes, we performed quantitative real-time PCR (qRT-PCR) testing. First, we investigated the interactions between collagen and the integrins family. qRT-PCR analysis revealed that NC-Col significantly upregulated the expression of *ITGA2* and *ITGA10* (Figure 4A). Specifically, a substantial upregulation of *ITGA2* expression was observed at all time points (24 h, 72 h, and 168 h) in the NC-Col group compared to the other groups. Similarly, *ITGA10* expression was also upregulated in the NC-Col group, especially at 24 and 168 h. In contrast, no significant differences were found between *ITGA5* and *ITGA11* expression across the groups; however, *ITGA5* expression in the control group was upregulated at 168 h compared to other groups.

Next, we focused on the *ACAN*, *SOX9*, and *COL II*, which are considered cartilage-specific genes, and *PRG4*, which regulates the production of lubricin. The results showed that the expression of *PRG4* was significantly upregulated, especially at 72 and 168 h. In addition, *ACAN*, *SOX9*, and *COL II* demonstrated different trends in the NC-Col groups compared to the other three groups (Figure 4B). For *ACAN*, there was no significant difference among the four groups. In contrast, *SOX9* expression in the NC-Col group was upregulated at 168 h compared to the control and the bovine-derived collagen matrix (BDC) group. Notably, although statistical significance was not observed at 24 h and 72 h, the results showed that *SOX9* expression was moderately increased in the NC-Col group. On the contrary, *COL2* expression in the control group was higher than that in the other three groups at all time points, with relatively stable expression of *COL1* across groups. Furthermore, there was no significant difference between the control and NC-Col groups for the value of *COL1*/*COL2* (an indicator of chondrocytes de-differentiation) at all time points. However, the value of *COL1*/*COL2* in the BDC and PDC groups was significantly higher than that in the control and NC-Col groups at 72 h (Figure 4C). These results indicate that, compared to PDC and BDC, NC-Col not only enhanced the expression of *PRG4* but also prevented the de-differentiation of human articular cartilage-derived chondrocytes at the same time.

### 2.3. RNA Sequencing Revealed Significant Differences Between NC-Col and PDC Groups in the Gene Expression Patterns of Chondrocytes

RNA sequencing (RNA-Seq) was conducted to investigate the difference in the effects of NC-Col and PDC on gene expression patterns of chondrocytes. Figure 5A shows the principal component analysis (PCA) of gene expression profiles in the NC-Col and PDC groups. The two groups are clearly separated along PC1 (35.8%) and PC2 (20.9%), indicating distinct transcriptional landscapes. The within-group clustering suggested high intra-group consistency, while the separation between the NC-Col and PDC groups reflected substantial differences in their global gene expression patterns. Compared to the PDC group, the NC-Col group exhibited significant transcriptional changes, with 359 genes upregulated (red) and 414 genes downregulated (blue) (Figure 5B). The first heatmap presented a hierarchical clustering analysis of all detected genes (Figure 5C). Overall, the two groups formed distinct clusters, indicating significant differences in gene expression patterns between them. In addition, the second heat map focused on the expression of specific genes of interest, including *SOX9*, *COL2A1*, and *PRG4*, some of which are cartilage-specific genes involved in chondrogenesis (Figure 5D). Several genes were significantly upregulated (red) in the NC-Col group, while they were primarily downregulated (blue) in the PDC group, suggesting regulatory changes in the NC-Col group during specific biological processes. According to Gene Ontology (GO) enrichment analysis, the differentially expressed genes (DEGs) showed enrichment for the specified terms in the biological process, cellular component, and molecular function modules (Figure 5E). It is noteworthy that the positive regulation of cell adhesion and negative regulation of cell migration were significantly enriched, further validating our experiments conducted above. In addition, Kyoto Encyclopaedia of Genes and Genomes (KEGG) pathway analysis indicated that the differential genes were enriched in pathways such as the PI3K-Akt signalling pathway, Ras signalling pathway, Notch signalling pathway, and WNT signalling pathway (Figure 5F). These results demonstrated that the effects of NC-Col on gene expression patterns of chondrocytes were significantly different from those of PDC, which represents a classical type of mammalian-derived collagen.

### 2.4. NC-Col Regulated ITGA10 and PRG4 Expression and Inhibited Chondrocyte Migration via the PI3K/Akt Signalling Pathway

Finally, western blotting (WB), qRT-PCR, and wound healing assay were performed to explore the mechanisms underlying the effects of the NC-Col on human articular cartilage-derived chondrocytes. By adding PI3K inhibitor LY294002 and Akt activator SC79, we confirmed that the PI3K/Akt signalling pathway was downregulated in the NC-Col group (Figure 6A). Furthermore, the qRT-PCR results showed that the expression of *ITGA10* in the four groups, from lowest to highest, was in the following order: the control group, the activator group, the NC-Col group, and the inhibitor group. Similarly, the expression of *PRG4* in the four groups, from lowest to highest, was in the following order: the activator group, the control group, the NC-Col group, and the inhibitor group (Figure 6B).

In the wound healing assay, the PI3K inhibitor LY294002 and the mTOR activator MYH1485 were used. Results showed that the migration rate in the four groups at 12 h, from lowest to highest, was in the following order: the inhibitor group (4.75% ± 1.29%, mean ± SD), the NC-Col group (8.14% ± 1.56%), the activator group (8.57% ± 2.28%), and the control group (12.54% ± 2.76%). At 24 h, the migration rate in the four groups, from lowest to highest, was in the following order: the inhibitor group (11.52% ± 1.83%, mean ± SD), the NC-Col group (18.97% ± 3.61%), the activator group (24.32% ± 6.37%), and the control group (30.23% ± 5.52%) (Figure 6C,D).

Collectively, our results showed that the PI3K/Akt signalling pathway was downregulated in the NC-Col group, and the upregulation of *ITGA10* and *PRG4* expressions was associated with the downregulation of the PI3K/Akt signalling pathway. In addition, it could be concluded that the NC-Col may suppress chondrocyte migration through inhibition of the PI3K/Akt/mTOR signalling pathway.

## 3. Discussion

The aim of this study was to evaluate the effects of the newly developed collagen NC-Col on human articular cartilage-derived chondrocytes in vitro. We first evaluated the effects of NC-Col on the proliferation, migration, and adhesion ability of human articular cartilage-derived chondrocytes. We found that NC-Col promoted the proliferation and adhesion ability of chondrocytes but inhibited their migration ability compared to the control and other collagen coating groups. It is in accordance with earlier findings that coating-collagen has a high ability to promote the proliferation of several types of cells, such as human fibroblasts, rat articular cartilage-derived chondrocytes, and human uterine leiomyosarcoma cells [9,20,21]. Moreover, it is also reported that compared to 3D collagen gel, collagen significantly promotes the proliferation ability of human articular cartilage-derived chondrocytes in 2D culture [8]. Therefore, despite the many advantages of 3D collagen gels in ACI, the significant proliferation-promoting effect of collagen matrix in 2D monolayer culture should not be overlooked.

Furthermore, our results showed that the adhesion ability of chondrocytes was significantly enhanced compared to the control and BDC groups (*p* < 0.001). This findingalso correlated well with the result that positive regulation of cell adhesion was significantly enriched in the DEGs (Figure 5F). Various receptor molecules, including integrins, cooperate to regulate cell adhesion. Integrin-mediated adhesion to the extracellular matrix remodels the cytoskeleton, promoting cell spreading and the clustering of integrins into focal adhesions [22]. Interestingly, KEGG pathway analysis in this study showed that focal adhesion signalling was significantly enriched (Figure 5E). Previous studies have shown that α1β1, α2β1, α10β1, and α11β1 are four collagen receptor integrins, and integrin α2β1 is the major receptor for type I collagen, which mediates adhesion to collagen via triggering several signalling pathways [23]. Moreover, Seung-Kye Cho et al. indicated that cell adhesion ability is promoted by upregulation of *ITGA2* [24]; Chao Liang et al. reported that *ITGA10* is also involved in the regulation of cell adhesion and migration [25]. Additionally, α5β1, which is also expressed in normal adult articular chondrocytes, serves as another cell adhesion receptor and mechanotransducer, mediating interactions between cells and the extracellular matrix [26]. Combining our results with those of higher *ITGA2* and *ITGA10* expression in the NC-Col collagen group, it can be concluded that NC-Col collagen mediates cell adhesion via α2β1 and α10β1, rather than α1β1, α5β1, or α11β1. Further studies are necessary to verify these findings. On the other hand, our results showed that the migration ability of chondrocytes was significantly inhibited in the NC-Col group. Previous studies have shown that the migration of endothelial cells and some cancer cells is regulated by the PI3K/Akt signalling pathway [27,28]. Combining our findings with the finding that the expression of *ITGA10* is regulated through the PI3K/Akt signalling pathway, it can be concluded that *ITGA10* is involved in the migration of chondrocytes via this pathway.

Cell adhesion encompasses both cadherin-mediated cell–cell adhesion and integrin-based cell–extracellular matrix (ECM) adhesion. In the present study, collective cell migration—the coordinated movement of cell groups—was investigated using the wound healing assay [29]. This process involves the integration of cell–cell and cell–ECM adhesions, enabling cells to maintain robust interactions with neighbouring cells and the underlying substrate [30]. Therefore, cell adhesion and migration are tightly correlated with complicated regulatory molecules. Our results showed that NC-Col promoted the adhesion ability of human articular cartilage-derived chondrocytes but inhibited their migration, which is in accordance with a previous study showing that the migration of human cancer cells could be inhibited through regulation of cell–matrix adhesion [31]. In addition, Juan Du et al. also found that draxin inhibits chick trunk neural crest migration by increasing cell adhesion [32]. However, some earlier findings indicated that the adhesion and migration ability of cells can be upregulated or downregulated at the same time [33,34]. The mechanism of NC-Col in relation to human chondrocyte migration and cell adhesion requires further investigation in the future.

Most importantly, ACI has become a well-established treatment for articular cartilage disorders of the knee. However, a persistent challenge remains in addressing the issue of chondrocyte de-differentiation during in vitro cell culture. The expression of cartilage-specific genes, such as *ACAN*, *SOX9*, and *COL II*, is down-regulated in de-differentiated chondrocytes [35]. A shift in the COL1/COL2 ratio has also been reported to be an indicator for de-differentiation [36]. On the other hand, lubricin, a product of the *PRG4* gene, is highly expressed in articular cartilage and plays a crucial role in boundary lubrication within articulating joints [37]. It prevents protein deposition from synovial fluid onto cartilage and serves as a protective factor against OA [37,38]. Our qRT-PCR results in the present study showed that *PRG4* and *SOX9* expression was upregulated in the NC-Col group. Additionally, the COL1/COL2 ratio remained comparable between the control and NC-Col groups but was significantly higher in the BDC and PDC groups at 72 h, indicating that the application of NC-Col in chondrocyte culture in vitro does not induce de-differentiation. It is worth noting that *SOX9* plays a crucial role in cartilage formation by promoting the production of collagen type II and aggrecan [9]. However, the observed downregulation of *COL II* expression and lack of significant upregulation of *ACAN* expression in the NC-Col group, compared to the control group, should be carefully considered. *SOX9* is widely recognised as a master regulatory transcription factor that directly activates the expression of key cartilage-specific genes, such as *COL2* and *ACAN* [9]. It is possible that the regulatory effects of NC-Col on *COL II* and *ACAN* expression occur at a later stage than its effect on *SOX9*. Therefore, further investigation is necessary to examine delayed gene expression responses and elucidate the underlying mechanisms. Conclusively, by combining our results, it is suggested that the newly developed collagen matrix NC-Col, used as a coating matrix, could better facilitate the growth of chondrocytes, maintain their phenotype, and upregulate *PRG4* expression in human articular cartilage-derived chondrocytes. This reveals the potential value for applying NC-Col collagen in ACI for the future treatment of OA.

There were some limitations to this study. First, only four patients (three females and one male) were included when abstracting human chondrocytes, resulting in an unequal distribution of genders (75% female and 25% male). Therefore, future research should increase the sample size and ensure gender balance to provide more reliable and comprehensive evidence. Second, human articular cartilage-derived chondrocytes were obtained from OA patients undergoing total knee replacement. Therefore, the results and conclusions obtained from this study are limited to pathological conditions of OA. Finally, this study was limited to in vitro cellular assays. Although the findings suggest that NC-Col has the potential to maintain the phenotype and upregulate *PRG4*—thereby providing a theoretical basis for its use as an adjunctive material in ACI for the treatment of OA—the clinical translational potential remains uncertain. To comprehensively evaluate and validate the clinical applicability of NC-Col, in vivo studies with appropriate animal models are imperative.

## 4. Materials and Methods

### 4.1. Human Samples and Ethics Statement

This experiment was conducted using cells derived from human articular cartilage and was approved by the Osaka University Institutional Ethical Committee (approval ID 23187-3). Written informed consent was obtained from all subjects, and all methods were performed in accordance with relevant guidelines and regulations. All patients were treated at Aihara Hospital, and sample collection was completed by the same person.

### 4.2. Materials

NC-Col, which is still a prototype in the development stage and has not yet been distributed on the market, was provided by Saraya Co., Ltd., Osaka, Japan. Commercially available PDC Type I-A (Lot. 230425, Nitta Gelatin, Osaka, Japan), BDC Aterocell IPC-30 (Lot. 124055, KOKEN, Tokyo, Japan), and FDC AQ-03A (Lot. 220801, TAKI CHEMICAL, Hyogo, Japan) were used. mTOR activator MHY1485 (cat. 5.00554) and PI3K inhibitor LY294004 (cat. 440202) were purchased from Calbiochem, San Diego, CA, USA. AKT activator SC79 (cat. 146428) was purchased from Abcam, Cambridge, UK. Six-well cell culture plates microplates with lids (cat. 3810-006N) were purchased from IWAKI, Shizuoka, Japan. Cell Counting Kit-8 reagent (cat. CK04) was purchased from DOJINDO, Kumamoto, Japan. The MULTISKAN (cat. 51119350) spectrophotometer was manufactured by Thermo Fisher Scientific, Waltham, MA, USA.

### 4.3. Cell Isolation and Culture

Knee articular cartilage was obtained from 3 female patients and 1 male patient aged between 75 and 82 years, who underwent total knee arthroplasty due to severe OA. Clinical data for donor patients included in this study are provided in Table 1. Chondrocyte isolation was performed as previously described [8]. Briefly, cartilage specimens were rinsed with phosphate-buffered saline (PBS), minced meticulously, and digested with 0.1% collagenase in Dulbecco Modified Eagle’s Medium/high glucose (DMEM, Cat. No. D6429, Sigma-Aldrich, Burlington, MA, USA) with 1% penicillin/streptomycin (Cat. No. 15140122, Thermo Fisher Scientific, Waltham, MA, USA) for 6 h at 37 °C. Cells were cultured in DMEM with 10% foetal bovine serum (FBS, Cat. No. 174012, Nichirei Bioscience, Tokyo, Japan) and 1% penicillin/streptomycin using commercial cell culture dishes or plates. Cells from passages two to four were used in this experiment.

### 4.4. Preparation of Collagen-Coated Plates and Dishes

Cell plates and dishes were coated with PDC, BDC, or FDC, following the manufacturers’ instructions. Cell plates and dishes were coated with NC-Col according to instructions from Saraya Co., Ltd. Briefly, NC-Col was diluted 10-fold with 0.01 N HCl, and 2 mL and 4 mL of the solution was added to a 100 mm culture dish and a 6-well cell plate, respectively. The dishes or plates were then incubated at room temperature for 60 min. After incubation, the solution was removed, and the dishes were rinsed three times with PBS. Coated dishes or plates were used immediately for experiments.

### 4.5. Cell Proliferation Assay

Chondrocytes were inoculated into 6-well cell culture plates coated with either NC-Col or PDC, or left uncoated as a control at densities of 1.5 × 10^5^, 1.0 × 10^5^, and 0.5 × 10^5^ cells/well and were cultured for 24, 72, and 120 h, respectively (*n* = 3 per group). Then chondrocytes were observed and imaged under an inverted microscope (DMi8; Leica Microsystems, Wetzlar, Germany). Finally, CCK-8 was added, and the optical density (OD) value was measured using a microplate reader at a wavelength of 450 nm.

### 4.6. Wound Healing Assay

To investigate the effects of NC-Col on the migration of chondrocytes, cells were inoculated into 6-well cell culture plates coated with either NC-Col, FDC, or PDC, or left uncoated as a control, at a density of 1.5 × 10^5^ cells/well (*n* = 3 per group). To investigate the involvement of the PI3K/Akt pathway in the migration ability of the cells, 2 μM mTOR activator MYH1485 and 10 μM PI3K inhibitor LY294002 were used. The concentrations were chosen based on previous studies [39,40]. Cells were inoculated into 6-well cell culture plates coated with either NC-Col or left uncoated as a control at a density of 1.5 × 10^5^ cells/well (*n* = 8 per group). Next, cell scratches were made. After washing the cell surface twice by adding PBS to remove cell debris, the cells were cultured for 12 and 24 h. Cells were observed and imaged using an inverted microscope (DMi8; Leica Microsystems, Germany) at 0 h, 12 h, and 24 h. Image analysis was performed using ImageJ with Fiji (version 1.54) [41]. The migration rate was calculated as follows: migration rate (%) = (initial wound area (t = 0 h) − residual area (t = 12 or 24 h))/initial wound area (t = 0 h) × 100%. Each experiment was conducted in triplicate.

### 4.7. Cell Adhesion Assay

A cell adhesion assay was performed as previously described with some modifications [42]. Chondrocytes were inoculated into 24-well plates (VIOLAMO; AS ONE, Osaka, Japan) coated with either NC-Col, BDC, or PDC, or left uncoated as a control, at a density of 5 × 10^4^ cells/well, and cultured for 20 min (*n* = 3 per group). After washing the cells 3 times with PBS, 50 μL/well of CCK-8 reagent was added. The OD value was measured after 90 min, using a microplate reader at a wavelength of 450 nm.

### 4.8. RNA Extraction and Quantitative Real-Time Polymerase Chain Reaction (RT-qPCR)

To investigate the effects of NC-Col on the migration of chondrocytes, cells were inoculated into 6-well plates coated with either NC-Col, BDC, or PDC, or left uncoated as a control at a density of 1.5 × 10^5^, 1.0 × 10^5^, 0.5 × 10^5^ cells/well and cultured for 24, 72, and 168 h, respectively (*n* = 3 per group). To investigate the involvement of the PI3K/Akt pathway in the gene expression of the cells, 10 μM Akt activator SC79 and 10 μM PI3K inhibitor LY294002 were used. The concentrations were chosen based on previous studies [39,43]. Chondrocytes were inoculated into 6-well plates coated with either NC-Col or left uncoated as a control at a density of 1.0 × 10^5^ cells/well and cultured for 72 h (*n* = 3 per group).

Total RNA from each specimen was extracted using Trizol (Cat. No. 15596026, Invitro gen; Thermo Fisher Scientific, USA) and a PureLink™ RNA Mini kit (Cat. No. 12183018A, Ambion; Thermo Fisher Scientific, USA). It was then reverse-transcribed into cDNA using a High-Capacity RNA-to-cDNA™ Kit (Cat. No. 4387406, Applied Biosystems; Thermo Fisher Scientific, USA). RT-qPCR was performed using Power SYBR™ Green Master Mix (Cat. No. 4367659, Thermo Fisher Scientific, USA) and the QuantStudio 7 Pro Real-Time PCR System (Thermo Fisher Scientific, USA), following the manufacturer’s protocol. The expression of gene glyceraldehyde-3-phosphate dehydrogenase (GAPDH) was used to normalise other genes, and the 2^−ΔΔCt^ method was used to analyse the relative expression of each gene. The primer sequences are listed in Table 2.

### 4.9. Construction and Sequencing of RNA-Seq Libraries

NC-Col was coated on 6-well cell culture plates. Chondrocytes in the control group were cultured on PDC-coated 6-well cell culture plates (Collagen Type I-coated MICROPLATE 6 Well with Lid; IWAKI, Japan). Chondrocytes were inoculated into both types of 6-well cell culture plates at a density of 0.5 × 10^5^ cells/well and cultured for 120 h. Then, total RNA was isolated using the Trizol reagent (Invitrogen; Thermo Fisher Scientific, Waltham, MA, USA) (*n* = 3 per group). Three tubes of samples from each group were sent for examination to ensure triplicate examination, with each tube containing total RNA from two wells to ensure adequate volume.

RNA integration was evaluated using the Agilent 2100 Bioanalyzer (Agilent Technologies, Santa Clara, CA, USA), and the effective concentration of RNA was subsequently quantified via qRT-PCR to ensure RNA quality. Illumina Nova Seq 6000 sequencing (San Diego, CA, USA) was performed after the quality control steps. All procedures, from RNA integration to RNA-Seq, were conducted by Rhelixa, Inc., Tokyo, Japan.

After RNA sequencing, DEGs were identified between the two groups. These genes were then subjected to GO functional enrichment analysis and KEGG pathway enrichment analysis using https://www.bioinformatics.com.cn (accessed on 10 October 2024), an online platform for data analysis and visualisation [44].

### 4.10. WB

To investigate the involvement of the PI3K/Akt pathway on the migration ability of chondrocytes, 10 μM Akt activator SC79 and 10 μM PI3K inhibitor LY294002 were used. Chondrocytes were inoculated into 6-well plates coated with either NC-Col or left uncoated as a control at a density of 1.5 × 10^5^ cells/well and cultured for 16 h. Total protein was extracted from the cells using RIPA buffer and was then separated by sodium dodecyl sulphate polyacrylamide gel electrophoresis (SDS-PAGE) (*n* = 3 per group). The separated proteins were transferred to polyvinylidene difluoride membranes, which were blocked with 5% non-fat milk dissolved in Tris-buffered saline containing 0.05% Tween 20 (TBS-T) for 1 h at room temperature. Membranes were then incubated overnight at 4 °C with primary antibodies specific for total Akt (1:1000; Cat. No. 9272, Cell Signalling, Danvers, MA, USA), p-Akt (1:2000, Cat. No. 9271, Cell Signalling), and GAPDH (1:1000, Cat. No. 2118, Cell Signalling). Finally, membranes were incubated for 1 h with horseradish peroxidase-conjugated anti-rabbit antibody (1:2000, Cytiva, Marlborough, MA, USA, Cat. No. 17640116) and visualised using an enhanced chemiluminescence system according to the manufacturer’s instructions.

### 4.11. Quantification and Statistical Analysis

Results from at least three independent experiments were obtained. Data are presented as the mean ± SD. Multiple comparisons were performed using one-way analysis of variance (ANOVA) followed by a post hoc Tukey–Kramer test. Statistical analyses were carried out using GraphPad Prism v9.5, and *p* < 0.05 was considered statistically significant.

## 5. Conclusions

The newly developed collagen matrix NC-Col, used as a coating matrix, could better enhance cell proliferation, maintain the phenotype, and upregulate the PRG4 expression of human articular cartilage-derived chondrocytes. The inhibition of the PI3K-Akt signaling pathway was found to be involved in some of these effects. Moreover, NC-Col collagen may serve as an effective matrix in ACI, offering new strategies for improving therapeutic outcomes in OA.

## Figures and Tables

**Figure 1 ijms-26-07826-f001:**
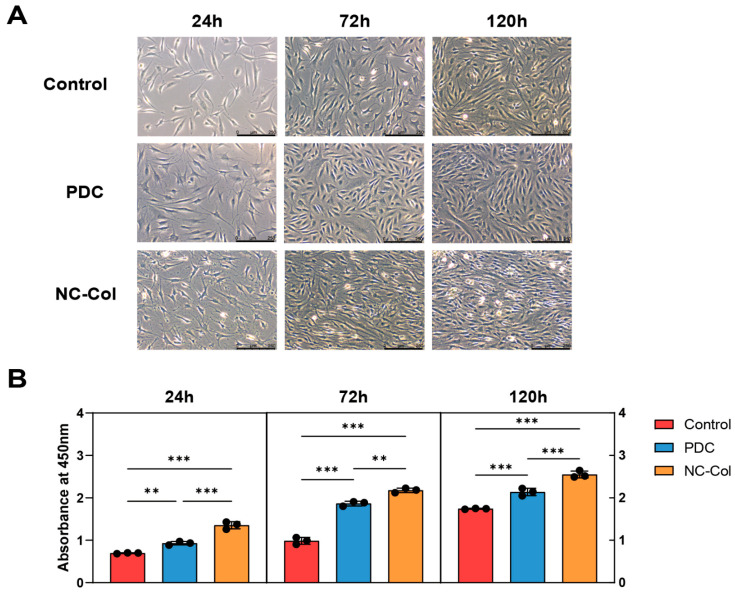
NC-Col promoted cell proliferation in chondrocytes. Proliferation ability was detected by CCK8. Chondrocytes were inoculated into 6-well cell culture plates coated with either NC-Col or PDC, or left uncoated as a control, and cultured for 24, 72, and 120 h. Each experiment was conducted in triplicate. (**A**) The picture is representative of 1 experiment of 3 performed. Magnification, 100×; scale bars: 250 μm. (**B**) OD values of three groups at 24, 72, and 120 h (*n* = 3 per group). Data are presented as the mean ± SD (error bars) and were analysed using one-way analysis of variance (ANOVA), followed by a post hoc Tukey–Kramer test. *p* < 0.05 was considered statistically significant. Statistical significance is indicated as follows: ** *p* < 0.01, *** *p* < 0.001. PDC, porcine-derived collagen matrix; OD, optical density.

**Figure 2 ijms-26-07826-f002:**
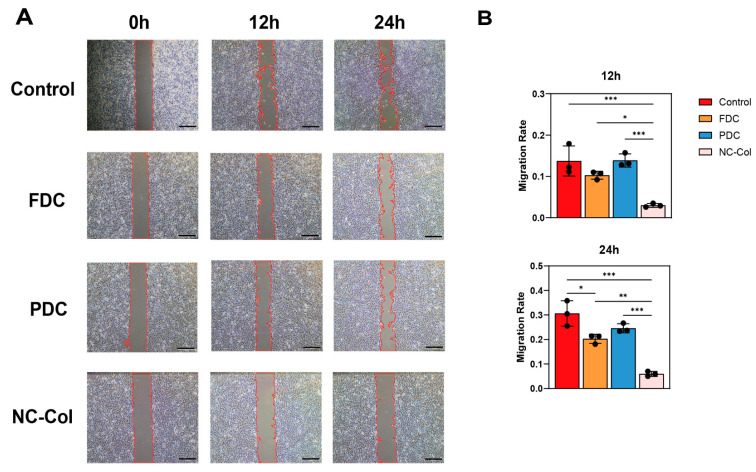
NC-Col inhibited the migration of chondrocytes. Migration ability was detected using a wound healing assay. Chondrocytes were inoculated into 6-well cell culture plates coated with NC-Col, FDC, or PDC, or left uncoated as a control, and cell scratches were made. Cells were observed and imaged at 0, 12, and 24 h. Each experiment was conducted in triplicate. (**A**) The picture is representative of 1 experiment of 3 performed. Cell front boundaries in the scratch area were outlined in red to indicate the wound edges. Magnification, 40×; scale bars: 500 μm. (**B**) Migration rates of 12 h and 24 h (*n* = 3 per group). Data are presented as the mean ± SD (error bars) and were analysed using one-way analysis of variance (ANOVA), followed by a post hoc Tukey–Kramer test. *p* < 0.05 was considered statistically significant. Statistical significance is indicated as follows: * *p* < 0.05, ** *p* < 0.01, *** *p* < 0.001. FDC, fish-derived collagen matrix; PDC, porcine-derived collagen matrix.

**Figure 3 ijms-26-07826-f003:**
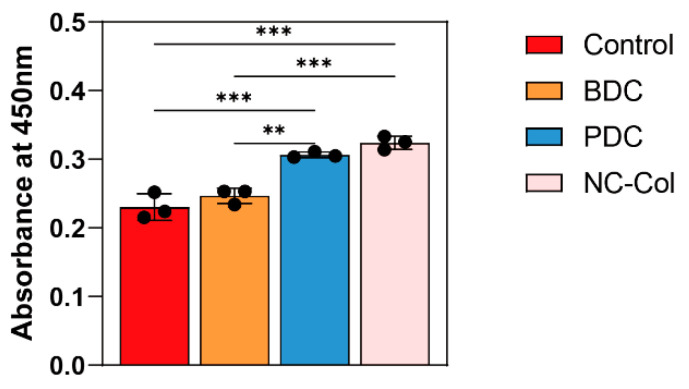
NC-Col enhanced the adhesion ability of chondrocytes. The cell adhesion ability was measured using a cell adhesion assay. Chondrocytes were seeded into 24-well plates coated with NC-Col, BDC, or PDC, or left uncoated as a control, and incubated for 20 min (*n* = 3 per group). The OD value was measured after the CCK-8 reagent was added at a wavelength of 450 nm. Each experiment was conducted in triplicate. Data are presented as the mean ± SD (error bars) and were analysed using one-way analysis of variance (ANOVA), followed by a post hoc Tukey–Kramer test. *p* < 0.05 was considered statistically significant. Statistical significance is indicated as follows: ** *p* < 0.01, *** *p* < 0.001. BDC, bovine-derived collagen matrix; PDC, porcine-derived collagen matrix. OD, optical density.

**Figure 4 ijms-26-07826-f004:**
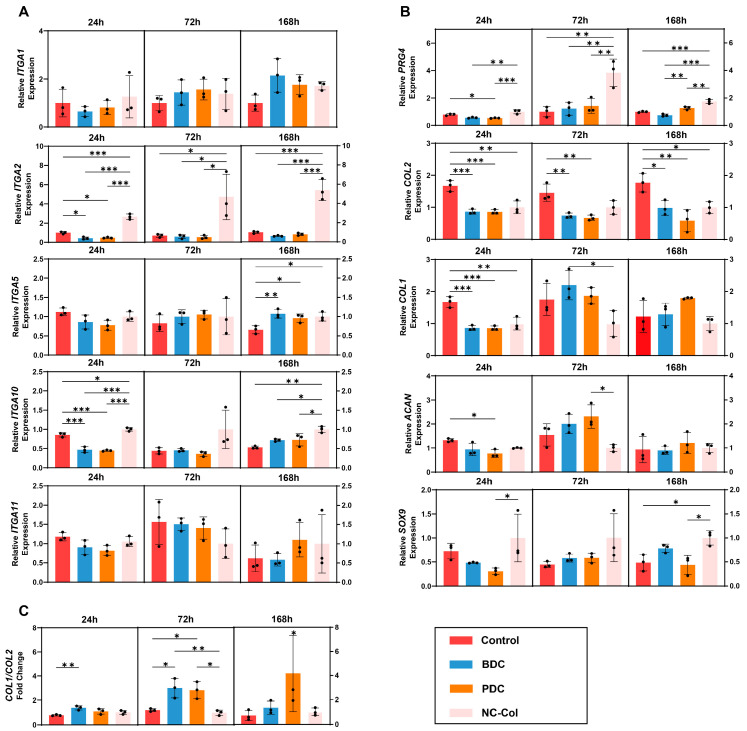
NC-Col upregulated the expression of *ITGA2*, *ITGA10*, and *PRG4* in chondrocytes. (**A**) The relative expression of *ITGA1*, *ITGA2*, *ITGA5*, *ITGA10*, and *ITGA11* in chondrocytes of the four groups at 24, 72, and 168 h (*n* = 3 per group). (**B**) The relative expression of cartilage-specific genes in chondrocytes of the four groups at 24, 72, and 168 h (*n* = 3 per group). (**C**) The ratio of COL1/COL2 in chondrocytes of the four groups at 24, 72, and 168 h (*n* = 3 per group). Each experiment was conducted in triplicate. Data are presented as the mean ± SD (error bars) and were analysed using one-way analysis of variance (ANOVA), followed by a post hoc Tukey–Kramer test. *p* < 0.05 was considered statistically significant. Statistical significance is indicated as follows: * *p* < 0.05, ** *p* < 0.01, *** *p* < 0.001. BDC, bovine-derived collagen matrix; PDC, porcine-derived collagen matrix.

**Figure 5 ijms-26-07826-f005:**
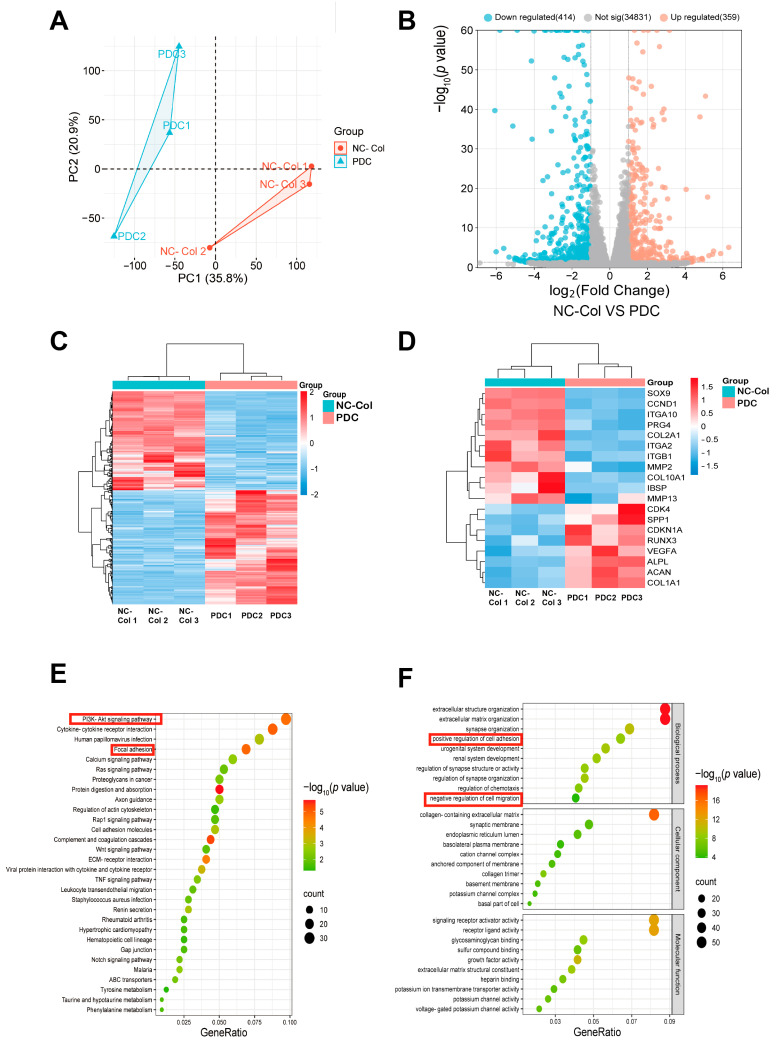
RNA-seq analysis of chondrocytes between NC-Col and PDC groups at 168 h (*n* = 3 per group). (**A**) Principal component analysis (PCA) scatter plot showing distinct separation between the NC-Col and the PDC groups based on principal components. (**B**) Volcano plot showing the distribution of differentially expressed genes between the NC-Col and the PDC groups. Red dots represent upregulated genes and blue dots represent downregulated genes. (**C**) Heatmap showing the differentially expressed genes between NC-Col and the PDC groups. Blue, lower expression; red, higher expression. (**D**) Heatmap showing the differentially expressed genes of interest between NC-Col and the PDC groups. Blue, lower expression; red, higher expression. (**E**) Dot plot of the GO pathway enrichment analysis. (**F**) Dot plot of the KEGG pathway enrichment analysis. (**E**,**F**) The horizontal axis represents the gene ratio, while the vertical axis represents the name of the enriched pathway. The colour scale indicates different thresholds of the *p*-value, and the size of the dot indicates the number of genes that correspond to each pathway. Pathways of interest are highlighted with red boxes. PDC, porcine-derived collagen matrix; GO, Gene Ontology; KEGG, Kyoto Encyclopaedia of Genes and Genomes.

**Figure 6 ijms-26-07826-f006:**
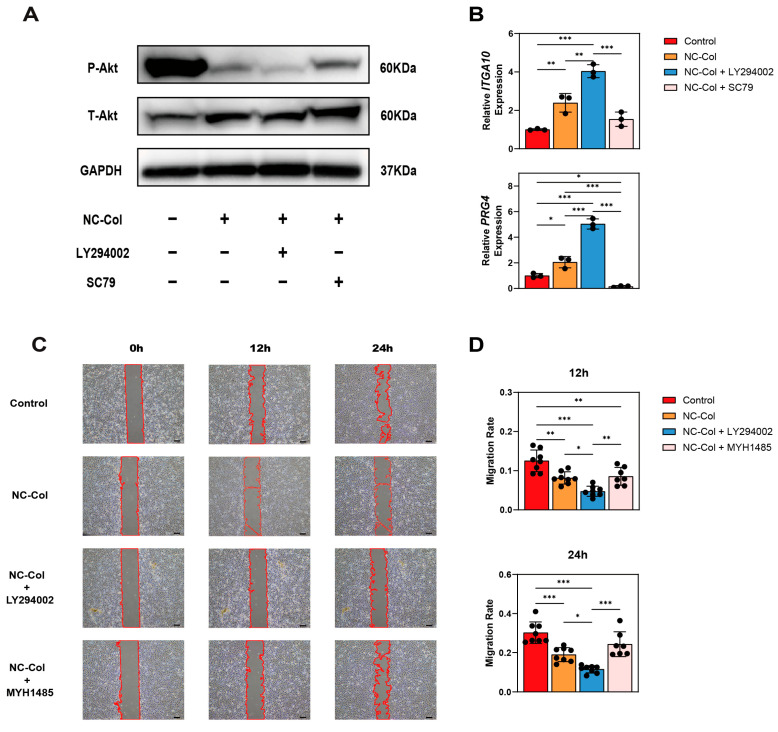
NC-Col regulated *ITGA10* and *PRG4* expression and inhibited chondrocyte migration via the PI3K/Akt signalling pathway. (**A**) PI3K/Akt signalling pathway was downregulated in the NC-Col group at 16 h (*n* = 3 per group). (**B**) Relative expression of *ITGA10* and *PRG4* at 72 h (*n* = 3 per group). (**C**) Migration ability was detected using a wound healing assay, and cells were observed and imaged at 0, 12, and 24 h (*n* = 8 per group). The picture is representative of 1 experiment of 8 performed. Cell front boundaries in the scratch area were outlined in red to indicate the wound edges. Magnification, 40×; scale bars: 200 μm. (**D**) The migration rates at 12 and 24 h (*n* = 8 per group). Data are presented as the mean ± SD (error bars) and were analysed using one-way analysis of variance (ANOVA) followed by a post hoc Tukey–Kramer test. *p* < 0.05 was considered statistically significant. Statistical significance is indicated as follows: * *p* < 0.05, ** *p* < 0.01, *** *p* < 0.001.

**Table 1 ijms-26-07826-t001:** Characteristics of donor patients.

No.	Age (Years)	Gender	Affected Knee	ICRS Grade Score of Cartilage Lesion *
1	78	Female	Left	4
2	75	Female	Right	3
3	80	Female	Left	3
4	82	Male	Left	4

* International Cartilage Repair Society (ICRS) Grade Score of cartilage lesion. https://cartilage.org/icrs-score-grade/ (accessed on 4 August 2025).

**Table 2 ijms-26-07826-t002:** qPCR primer sequences for target genes.

Primers	Forward (5′—3′)	Reverse (5′—3′)
GAPDH	TCTCTGCTCCTCCTGTTCGAC	GTTGACTCCGACCTTCACCTTC
ITGA1	CAGCCCCACATTTCAAGTCGT	ACCTGTGTCTGTTTAGGACCA
ITGA2	GCAACTGGTTACTGGTTGGTT	GAGGCTCATGTTGGTTTTCATCT
ITGA5	GCCTGTGGAGTACAAGTCCTT	AATTCGGGTGAAGTTATCTGTGG
ITGA10	CTTCAGTTCTGGGATATGTGCC	CCAGTCTTCGTAGGAAGGTCT
ITGA11	TCACGGACACCTTCAACATGG	CCAGCCACTTATTGCCACTGA
SOX9	AGGCAAGCAAAGGAGATGAA	TGGTGTTCTGAGAGGCACAG
ACAN	ACAGCTGGGGACATTAGTGG	GTGGAATGCAGAGGTGGTTT
PRG4	TCCATTCAGTCCACCATCTCC	TGTCCAGTTATCCTCCAAATCCT
COL1A1	GACCAGCAGACTGGCAACCT	GCTGAGGTGAAGCGGCTGT
COL2A1	TGGACGCCATGAAGGTTTTCT	TGGGAGCCAGATTGTCATCTC

## Data Availability

All data generated or analysed during this study are included in this article. Further enquiries can be directed to the corresponding author.

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
