# Peer review of "Effects of a Novel Mammalian-Derived Collagen Matrix on Human Articular Cartilage-Derived Chondrocytes from Osteoarthritis Patients"

_ijms, 2025, doi:10.3390/ijms26167826_

Round 1

Reviewer 1 Report

Comments and Suggestions for Authors

This manuscript explores the in vitro effects of a newly developed mammalian-derived collagen matrix (NC-Col) on OA chondrocytes, comparing it with other collagen matrices. 

The data appear solid and support the conclusions, only with minor recommendations:

  1. Although SOX9 and PRG4 are mildly upregulated, COL2A1 is consistently lower in the NC-Col group compared to control, which contradicts claims of phenotype maintenance. Consider adding short a discussion about this inconsistency.
  2. Some figures lack clear labels for conditions. Indicate sample size (n), statistical significance indicators, and repeatability. Adding it to the description should be sufficient and easier to edit, if possible. 
  3. Try to clearly distinguish correlation from causation in mechanistic claims. 
  4. Please check, if the references are cited according to the journal template

After minor changes I can recommend this article for acceptance.

Author Response

  • Comments 1: Although SOX9 and PRG4 are mildly upregulated, COL2A1 is consistently lower in the NC-Col group compared to control, which contradicts claims of phenotype maintenance. Consider adding short a discussion about this inconsistency.

Response 1: Thank you for pointing this out. We agree with this comment. Therefore, we have added a short description in the discussion about the inconsistency you mentioned (Page 10, starting from line 329).

  • Comments 2. Some figures lack clear labels for conditions. Indicate sample size (n), statistical significance indicators, and repeatability. Adding it to the description should be sufficient and easier to edit, if possible.

Response 2: We agree with this comment. Firstly, we have revised Figure 2B. In the original version, the label for PDC was mistakenly shown as BDC. Then, we revised all of the captions of the figures, adding the information of sample size (n), statistical significance indicators, and repeatability.

  • Comments 3. Try to clearly distinguish correlation from causation in mechanistic claims.

Response 3: Thank you for pointing this out. We agree with this comment. Therefore, we revised the last paragraph of Results 2.4(starting from Page 8 Line 246).

  • Comments 4. Please check, if the references are cited according to the journal template

Response 4: Thank you for pointing this out. All of the references were revised according to the journal template.

Reviewer 2 Report

Comments and Suggestions for Authors

The authors of this article have investigated the use of a mammalian-derived collagen matrix (N-COL) for autologous chondrocyte implantation (ACI). Using articular cartilage chondrocytes from total knee arthroplasty patients, the authors demonstrated that upon culture on this N-COL matrix was able to redifferentiate chondrocytes, expressing SOX9, COL2A1, ACAN, to an enhanced expression compared to other animal derived matrices. Furthermore, integrin expression, particularly ITA10 and ITA2 were upregulated on this matrix, alongside PRG4. Adhesion was enhanced on this particular matrix with migration inhibited on N-COL matrix. This latter property was under the control of the PI3K/Akt pathway.

The results of the investigation demonstrate the promise of this matrix for MACT. The authors should consider the following points

  1. There are inconsistencies in the figure with respect to the use of fish, porcine and bovine-derived collagen matrix in figures 1 and 2. Please present the data the porcine and bovine data for all the figures to ensure consistency with each figure, specifically figure 1 and 2.
  2. What are the known differences in collagens between porcine, bovine and human collagens in respect to articular cartilage content. Could this have influenced the chondrocyte redifferentiation and their gene expression ?
  3. What focal cartilage type would be suitable for the proposed treatment using the N-COL matrix ? This is in respect to traumatic and early OA focal lesions and the success of this treatment in the long-term. Could this matrix help to improve outcomes ?
  4. What was the control used for the RNAseq studies ? Normally, a healthy chondrocyte cohort (non-OA) is nominally used for this study to understand the improvement in the use of these matrices.
  5. Why was this not performed in a 3D scaffold rather than on coatings as per the manuscript ? Would this not be more appropriate for understanding how this scaffold would work in a translational approach.
  6. What was the Kellgren-Lawrence or ICRS cartilage score for patients used in this study ? This should be stated in the materials and methods section.

Author Response

  • Comments 1. There are inconsistencies in the figure with respect to the use of fish, porcine and bovine-derived collagen matrix in figures 1 and 2. Please present the data the porcine and bovine data for all the figures to ensure consistency with each figure, specifically figure 1 and 2.

Response 1: Thank you for pointing this out. We agree with this comment and have revised Figure 2B accordingly. In the original version, the label for PDC was mistakenly shown as BDC. However, in the proliferation assay (Figure 1), we only included three groups and did not set up a BDC group. Broadly speaking, both PDC and BDC refer to traditional, commercially available, and widely used mammalian-derived collagen matrices. The aim of our study is to compare the NC-Col matrix with commonly used mammalian-derived collagen matrices, such as PDC and BDC.

  • Comments 2. What are the known differences in collagens between porcine, bovine and human collagens in respect to articular cartilage content. Could this have influenced the chondrocyte redifferentiation and their gene expression?

Response 2: Thank you for pointing this out. According to previous studies and published literature, there are relatively few studies focusing on the differential effects of collagen matrices derived from various species. However, some studies have explored the impact of different types of collagens on articular cartilage chondrocytes. Therefore, in our study, we aim to compare the effects of collagen matrices resourced from different species on human articular cartilage-derived chondrocytes.

  • Comments 3. What focal cartilage type would be suitable for the proposed treatment using the N-COL matrix? This is in respect to traumatic and early OA focal lesions and the success of this treatment in the long-term. Could this matrix help to improve outcomes?

Response 3: We appreciate your insightful question. Based on the findings of our in vitro study, the NC-Col matrix demonstrated favorable effects on human articular chondrocytes derived from OA patients. While our current results are limited to in vitro experiments, it is only a hint that the novel material may have the potential to expand the application field of ACI to treat OA patients (considering ACI is not promoted as a standard treatment for OA).

Moreover, ACI is an established and well-accepted procedure for the treatment of localized full-thickness cartilage defects of the knee. Generally speaking, compared to Osteochondral transplantation (OCT)—including OATS (Osteochondral Autograft Transfer System) and mosaicplasty, there is an indication for ACI for defects greater than three to four cm2 and as a second-line treatment for smaller defects. For athletically active and younger patients, ACI for defects greater than 2.5 cm2 is recommended. In the context of treatment of traumatic and early OA focal lesions with NC-Col, further in vivo studies and long-term evaluations are required.

  • Comments 4. What was the control used for the RNA-seq studies? Normally, a healthy chondrocyte cohort (non-OA) is nominally used for this study to understand the improvement in the use of these matrices.

Response 4: We thank the reviewer for this valuable comment. In our RNA-seq experiment, we did not use a healthy chondrocyte cohort (non-OA) as a control. Instead, chondrocytes cultured on a porcine-derived collagen (PDC) matrix was used as the control group. Our primary objective was to compare the effects of a novel collagen matrix (NC-Col) with those of a traditionally commercialized, mammalian-derived collagen matrix (PDC) on human articular chondrocytes derived from OA patients.

We fully acknowledge that including a healthy chondrocyte cohort would provide a more comprehensive understanding of the therapeutic potential of NC-Col. However, obtaining human chondrocytes from healthy individuals poses significant ethical and practical challenges. Nonetheless, we appreciate your suggestion, and incorporating such a comparison will be strongly considered in future studies, possibly through the use of induced pluripotent stem cell-derived chondrocytes or suitable animal models.

  • Comments 5. Why was this not performed in a 3D scaffold rather than on coatings as per the manuscript? Would this not be more appropriate for understanding how this scaffold would work in a translational approach.

Response 5: We appreciate the reviewer’s insightful comment regarding the use of 3D scaffolds. In this study, we employed NC-Col in a coating format because this material is still a prototype currently under development and has not yet been commercially distributed. The NC-Col was kindly provided by Saraya Co., Ltd., and at this stage, a 3D scaffold form of this material has not yet been established. Nevertheless, our in vitro experiments demonstrated the promising potential of NC-Col in promoting chondrocyte proliferation and regulating cartilage specific gene expression. Based on these encouraging results, further in vitro studies using a 3D scaffold format of NC-Col may be considered in future investigations. This initial work provides foundational data necessary for the development and optimization of a future 3D scaffold format of NC-Col.

  • Comments 6. What was the Kellgren-Lawrence or ICRS cartilage score for patients used in this study? This should be stated in the materials and methods section.

Response 6: Thank you for pointing this out. We agree with this comment and have added the ICRS grade score for the cartilage lesions of the patients included in this study (see Page 11, Table 1).